# The Role of Interstitial Fluid Pressure in Cerebral Porous Biomaterial Integration

**DOI:** 10.3390/brainsci12040417

**Published:** 2022-03-22

**Authors:** Fabien Bonini, Sébastien Mosser, Flavio Maurizio Mor, Anissa Boutabla, Patrick Burch, Amélie Béduer, Adrien Roux, Thomas Braschler

**Affiliations:** 1Department of Pathology and Immunology, Faculty of Medicine, University of Geneva, Rue Michel-Servet 1, CH-1022 Geneva, Switzerland; fabien.bonini@unige.ch (F.B.); anissa.boutabla@hcuge.ch (A.B.); amelie.beduer@volumina-medical.ch (A.B.); 2Neurix SA, Avenue de la Roseraie 64, CH-1022 Geneva, Switzerland; seb.mosser@gmail.com; 3Haute École du Paysage, d’Ingénierie et d’Architecture de Genève, Haute École Spécialisée de Suisse Occidentale (HEPIA HES-SO), University of Applied Sciences and Arts Western Switzerland, CH-1202 Geneva, Switzerland; fmor82@gmail.com (F.M.M.); adrien.roux@hesge.ch (A.R.); 4Volumina-Medical SA, Route de la Corniche 5, CH-1066 Epalinges, Switzerland; Patrick.Burch@geistlich.com

**Keywords:** glymphatic system, interstitial fluid, biointegration, biomaterial

## Abstract

Recent advances in biomaterials offer new possibilities for brain tissue reconstruction. Biocompatibility, provision of cell adhesion motives and mechanical properties are among the present main design criteria. We here propose a radically new and potentially major element determining biointegration of porous biomaterials: the favorable effect of interstitial fluid pressure (IFP). The force applied by the lymphatic system through the interstitial fluid pressure on biomaterial integration has mostly been neglected so far. We hypothesize it has the potential to force 3D biointegration of porous biomaterials. In this study, we develop a capillary hydrostatic device to apply controlled in vitro interstitial fluid pressure and study its effect during 3D tissue culture. We find that the IFP is a key player in porous biomaterial tissue integration, at physiological IFP levels, surpassing the known effect of cell adhesion motives. Spontaneous electrical activity indicates that the culture conditions are not harmful for the cells. Our work identifies interstitial fluid pressure at physiological negative values as a potential main driver for tissue integration into porous biomaterials. We anticipate that controlling the IFP level could narrow the gap between in vivo and in vitro and therefore decrease the need for animal screening in biomaterial design.

## 1. Introduction

Neurological disorders are a major cause of death and disability worldwide and have been a growing social and economic burden for the past 20 years [1,2,3]. Despite the existence of stem cells in the adult brain parenchyma [4,5], the brain has a poor self-regenerative potential [6,7]. As a consequence, severe damage to the brain tissue is largely irreversible, and often gives rise to major neurological deficits reflecting the location and size of the injured brain area [8]. Recent advances in regenerative medicine have shown promise in management of central nervous system diseases such as stroke [9,10] or Parkinson’s disease [11,12]. Regenerative techniques have taken advantage of the use of biomaterials to deliver cells and/or drugs to increase neuroplasticity or to replace defective cells [13,14,15]. In neuronal cell transplantation, biomaterials provide an adequate physical support and growth environment, enhancing graft survival, localization and function [16,17,18,19].

Beyond the act of viable cell transfer, the next challenge in biomaterial-assisted cell transplantation is the understanding and optimization of the mechanisms of integration of the implants with the host tissue, a process also referred to as biointegration [20]. The host response inevitably comprises early inflammatory events, leading to immigration of host immune cells [21,22]. However, in a desired regenerative course, endogenous stem cells and possibly transplanted stem cells are thought to then ensure de novo tissue formation and functional maturation [23]. In addition, 3D tissue ingrowth from the surrounding environment occurs as well, providing vascularization, but also stromal and possibly functional components [24,25]. Vascular support is of particular importance in the context of neuronal transplantation for which cells are metabolically demanding and thus requires a rapid access to oxygen and nutrients upon transplantation [26].

The traditional approach to both cell delivery and fostering biointegration is the inclusion of bioactive molecules such as extracellular matrix (ECM) proteins (e.g., laminin [27] or collagen [28]) or peptides adhesion motives (e.g., RGD [29] or IKVAV [30]) into biomaterial scaffolds. In parallel, slow release of growth factors, small molecules, chemokines or other cell-attractive molecules is often used to enhance the cell migration into the biomaterial structure [31]. It is then by active cell migration that current biomaterials achieve tissue colonization [32]. In this case, cell-biomaterial interaction as well as the cellular response to individual substances can be screened in vitro by available, relatively simple techniques [33,34].

We hypothesize here that biomechanical events complete or even dominate biological signaling and cell-autonomous migration in neural biomaterial integration, particularly in porous materials. Indeed, beyond the cellular events, implantation of a biomaterial can also be expected to change the local tissue mechanics. For a porous material, two types of forces will be exchanged with the local tissue [35]: (i) the solid tissue pressure is the accumulation of the forces applied by the solid elements of the tissue onto the solid part of the biomaterial, (ii) the tissue interstitial fluid will directly communicate with the pore fluid of the biomaterial, transmitting the interstitial fluid pressure (IFP) to the pore fluid.

Positive solid tissue pressure forces an intimate contact between the surrounding tissue and the biomaterial implant. This makes appropriate modulation of the biomaterial stiffness essential [36]. Indeed, stiffer biomaterials create a mechanical mismatch with the host tissue that is more likely to trigger chronic inflammation, gliosis and encapsulation [37,38,39,40]. On the other hand, softer biomaterials present poor stability and are not capable of withstanding solid tissue pressure [41,42]. Taken together, they represent the physiological mechanical constraints of the tissue. Alongside biocompatibility and biodegradability, biomaterial stiffness is therefore a mandatory design consideration [43].

In contrast, little is known about the impact of the IFP on biomaterial tissue integration. It is therefore our aim to study the role of IFP in neural tissue integration. Our specific hypothesis is that the interstitial fluid pressure, which is normally slightly below ambient atmospheric pressure [35,44], is a key player in biointegration by physically driving bulk tissue invasion. We investigate the influence of the IFP on neural biointegration with the aid of capillary microfluidic system that mimics the workings of the glymphatic system, which physiologically regulates the IFP in the brain [45].

The glymphatic system indeed regulates interstitial fluid renewal and pressure by filtration and re-uptake, as it happens in most other tissues [45,46,47,48]. In addition, it presents anatomic particularities that constitute the blood–brain barrier. In the glymphatic system, cerebrospinal fluid circulates in paravascular conduits, driven by arterial pulsatility and respiration [45]. It is filtrated from the arterial part of this system into the proper neural interstitial space. This filtration occurs through the astroglial water channel aquaporin-4, limiting brain access for hydrophilic molecules to a very small size [49]. Re-uptake occurs from the venous part of the paravascular system [45]. Under physiological conditions, the cerebral IFP is below atmospheric [35,44]. This results in negative values approximately equal to the positive solid tissue pressure, therefore promoting tissue compaction [35].

Interstitial fluid pressure cannot be controlled in cell culture flask and dishes. In insert cultures, hydrostatic level differences on the order of a few millimeters of water column are used to maintain air–liquid interface cultures, [50] remaining, however, far below physiological intracerebral values [44] in magnitude. More deeply, subatmospheric pressure is used for clinical enhancement of chronic wound healing in “negative-pressure wound therapy” [51,52] and corresponding efforts to apply strongly negative IFP to in-vitro cultures have been made using bioreactors employing negative pressure wound dressings [53]. However, this approach implies direct contact of the negative pressure dressings and the 3D culture, leading to coupling between negative pressure, mechanical forces and interstitial flow [53].

Inspired by the simple yet elegant fluidic architecture of the glymphatic system, we propose here the design of a brain-on-a-chip capillary microfluidic system instead. This system recapitulates the role of the glymphatic system by applying controlled negative IFP up to the physiological range to long-term 3D organoid CellBrain cultures (commercially available, Neurix). The negative pressure is transmitted to the 3D culture by capillary contiguity through the filter membrane of cell culture inserts used to support the 3D culture [50]. This avoids direct contact between capillary conductor and the 3D culture, and decouples IFP from fluid flow. In order to satisfy the known high metabolic and gas exchange demands of neural tissue, we implement the device using an air–liquid interface geometry [50,54]. We use this setup to investigate in vitro the role of the IFP on the biointegration of an established highly porous and biocompatible carboxylmethylcellulose (CMC) biomaterial scaffolds [25,55] with metabolically demanding neuro-organoids.

The aim is to understand whether the interstitial fluid pressure could have role in biomaterial integration, and if so, to better understand the order of magnitude as compared to the known favorable effects of cell adhesion motives [20]. We further aim at outlining biomaterial properties that would best exploit interstitial fluid pressure-driven biointegration, as well as the possible mechanisms involved. Finally, the device is optionally compatible with microelectrode recording for functional assessment. Considering the potential of this biomaterial for cell transplantation and cerebral tissue reconstruction, we investigated the impact of these culture conditions on the neuronal spontaneous activity and graft function.

## 2. Materials and Methods

### 2.1. Biomaterial Manufacturing

We based the synthesis of our porous biomaterial on cryogelation of carboxymethylcellulose, followed by forceful fragmentation of the bulk material to obtain irregular and porous particles as described previously [25,55]. Briefly, a reactive premix was prepared by adding 13.65 g of carboxylmethylcellulose (Aqualon^TM^, 90.5 KDa, DS: 0.84, Ashland, Wilmington, DE, USA, ref 891158), 6.30 g of PIPES buffer (1,4-Piperazinediethanesulfonic acid, Sigma-Aldrich, St. Louis, MO, USA, P6757), 486 mg of adipic acid dihydrazide (ADH, Sigma, Kawasaki, Japan, A0638) and NaOH pellets (1.2 g, Carlo Erba, Val-de-Reuil, Germany, 480507) and the total volume was adjusted to 1000 mL with deionized water. In order to better visualize the biomaterial, 6-aminofluorescein was added to the reactive premix (final concentration 10 µM). Then, EDC (1-ethyl-3-(-(3-dimethylaminopropyl) carbodiimide, Sigma-Aldrich, E1769) was added (4 mg/mL of reactive premix) and after mixing the solution was then filled into 50 mL falcon tubes and left overnight in the freezer set to −20 °C.

Porous biomaterials were thawed and washed with deionized water and PBS and then fragmented [25]. The bulk material was roughly cut using a razor blade and filled into a 10 mL syringe and then passed sequentially through 18 G, 20 G and 22 G needles. After autoclaving, the biomaterials were optionally post-modified with Matrigel (Engelbreth-Holm-Swarm murine sarcoma basement membrane, Sigma-Aldrich, Switzerland, E1270-1ML), laminin (Sigma, L2020) or collagen type I (Sigma-Aldrich, C4243).

### 2.2. Protein Covalent Immobilization: Matrigel and Laminin

Free carboxyl groups found on the carboxylmethylcellulose polymer strands allow for facile covalent attachment of different adhesion molecules with primary amines, using conjugation chemistry based on carbodiimide-initiated amidation [56]. Accordingly, the biomaterial was modified with Matrigel and laminin according to established protocols, with minor modifications [57]. Briefly, the biomaterial was dried on a filtration unit (TPP, Switzerland, 191060), washed and incubated in a solution of either Matrigel (Sigma-Aldrich, E1270) or laminin (Sigma-Aldrich, L2020) at 10% mass ratio (i.e., 1:10, protein mass/CMC mass) diluted in MES (Morpholineethanesulfonic acid, Sigma, M3671) 50 mM, adjusted to pH = 5.5 with NaOH, overnight in an incubator at 37 °C. The day after, the biomaterial was collected, dried on a filtration unit and incubated for 15 min in a solution of the water-soluble carbodiimide EDC at 1 mg/mL in MES buffer 50 mM (pH = 5.5). After abundant washing the biomaterial was kept in PBS at 4 °C.

For Matrigel labelling with rhodamine isothiocyanate (RITC, Sigma-Aldrich, 283924-100MG), a stock solution of RITC was prepared at 10 mg/mL in isopropanol. Furthermore, a basic buffer was prepared by dissolving 42 g of sodium bicarbonate and 6.4 g of NaOH in a final volume of 500 mL, resulting in a measured pH of 9.5. The reaction was then initiated by mixing 1 mL of Matrigel solution, 100 microliters of the carbonate buffer, 250 μL of pure isopropanol and 50 microliters of the 10 mg/mL RITC solution in isopropanol. After 24 h of incubation in the cold, this solution was used without further purification for visualization experiments as little reaction of leftover RITC with the scaffolds is expected.

### 2.3. Protein Covalent Immobilization: Collagen Type I

Covalent modification with collagen I followed established protocols [25]. Briefly, the biomaterial was dried on a filtration unit (TPP, Switzerland, 191060), washed and incubated in an acetic acid buffer 0.5 M (pH = 4). Then, the biomaterial was immerged in a solution of collagen type I at 10% (Sigma-Aldrich, Switzerland, C4243) before covalent crosslinking using a solution of EDC at 1 mg/mL in MES buffer 50 mM (pH = 5.5). After abundant washing the biomaterial was kept in PBS at 4 °C.

### 2.4. Bradford Protein Quantification Assay

To characterize protein immobilization and stability on biomaterial preparations, we quantified the amount of protein remaining on the biomaterials after 24 h of incubation at 37 °C in DMEM. For this, commercial stock solution of trypsin (Gibco, Switzerland, 25300-054) was diluted in deionized water at 1:12 (*v*/*v*, final trypsin concentration at 0.2 mg/mL). Around 5–30 mg of dry mass of unmodified (EDC treatment only) or Matrigel modified biomaterial were dehydrated on a filtration unit (TPP, Switzerland, 191060), washed twice with deionized water and incubated in the diluted trypsin solution at 37 °C for 2 h. In addition, serial dilution of Matrigel (stock at 10 mg/mL) ranging from 5 mg/mL to 0.0625 mg/mL was prepared in the same conditions for determination of a standard curve.

The supernatant was then segregated from the biomaterial using a cell strainer (Greiner, Kremsmünster, Austria, 40 µm, 542,040) placed in a 50 mL falcon tube and centrifuged for 30 s a 3000 rpm. 10 μL of the recovered protein content (either dilutions or the flow through) were then added to 200 μL of working solution of Bradford reagent (1:3 *v*/*v* of the stock solution into deionized water, Bio-Rad, Hercules, CA, USA, 5000-0006) according to the supplier guideline. The plate was briefly vortexed for 10 s at 800 rpm to remove air bubbles. The plate was then incubated at RT for 30 min and read using a SpectraMax Paradigm firmware V1.2 b103 and driven by SoftMax Pro v7.1.0 build 246,936. Background absorption was estimated using deionized water and subtracted from all data.

### 2.5. Glymphatic Brain-on-a-Chip Pressure Device

The glymphatic brain-on-a-chip device was prototyped using 21,000 µL tips boxes (Axygen scientific, Santa Clara Valley, CA, USA, TF-1005-WB-R-S) fused together using a soldering iron and sealed with silicone glue. For the functional unit, a hole was made into the reservoir tube (50 mL falcon tube) corresponding to the desired pressure (namely 3 cm, 6 cm or 9 cm from the CellBrain platform). Then a 15 mL falcon tube was melted and attached to the reservoir and the collection tube using a heat gun. A plastic grid was then added on the top of the functional unit. On the bottom of both collection and reservoir tube, a small hole was drilled and a 200 μL tips was then attached. Both tubes were then connected using silicone tube with an inner diameter of 0.5 mm. The tube was then plugged to the pump (Boading Shenchen precision pump, China, OEM-MZ20/minipump01 powered by a DC motor, motor flow rate 1.39 +/− 0.01 mL/min). Importantly, the components of this device are polypropylene-based to withstand sterilization by autoclaving.

Validation of the correspondence between set and actual pressure values was conducted using both a micropipette technique and a digital pressure sensor (Adafruit BMP280 sensor, Art.Nr. 301-29-199, Distrelec, Switzerland). For the micropipette approach, a microcapillary (152 mm length, 0.75 mm inner diameter, World Precision Instrument, Germany, TW100-6) was first calibrated by measuring the capillary forces in the absence of external negative pressure (0 kPa). To do so, the capillary was applied at the very surface of a tube filled with deionized water. After 10 sec, the capillary was removed, and the height reached by the deionized water inside the capillary was measured using a ruler (1 cm H_2_O = 0.098 kPa). The measurement was then repeated while contacting the cloth of each functional unit. The difference in height was calculated to give the measurement of the pressure in each functional unit. To note, the capillary forces of these microcapillaries allowed us to measure negative pressures up to −0.5 kPa; for more negative values, the capillary force provided by the micropipettes is insufficient and no more fluid enters the capillary. The dataset was therefore extended (and cross-validated) using digital pressure measurement. For this, we measured the equilibrium barometric pressure in a closed measurement cavity on top of a water column of known height in capillary contact with the cloth of the functional units using the BMP280 digital pressure sensor (interfacing via an Arduino Uno, sketch slightly modified from the example script for the BMP280, and provided in the raw data repository, using the built-in serial monitor in the Arduino development environment). We compared this to the barometric pressure in the free atmosphere measured by the same sensor with a time difference of at most two minutes between the measurements to avoid influence of atmospheric pressure drift (the BMP280 is sensitive to pressure changes in the Pa range). Correcting for the known water column height used, this yields the capillary pressure by difference.

### 2.6. LUHMES-Based CellBrain Production

Brain tissue organoids (commercial name: CellBrain) were prepared by Neurix SA, Geneva, Switzerland. The CellBrain organoids are grown from wild-type LUHMES cells (ATCC^®^ CRL_2927™) as previously described [58]. Briefly, proliferation medium was prepared with Advanced DMEM/F12 (Gibco, Life Technologies, Carlsbad, CA, USA, 12634010) supplemented with 2 mM l-Glutamine (Sigma-Aldrich, G7513), 1× N2 (Gibco, Switzerland, 17502948) and 0.04 μg/mL recombinant basic fibroblast growth factor (bFGF, R&D Systems, UK, 3718-FB-025). Differentiation medium was prepared with Advanced DMEM/F12 containing 2 mM l-Glutamine, 1× N2 supplement, 1 mM dibutyryl cAMP (Santa Cruz, Germany, sc-201567), 2 μg/mL tetracycline (Sigma-Aldrich, 87128), and 2 ng/mL recombinant human glial cell line-derived neurotrophic factor (GDNF, Geminibio, CA, USA, 300-121P). Four million cells were seeded in one Nunclon™ (Nunc) T75 flask pre-coated for 3 h with 50 μg/mL poly-L-ornithine (PLO, Sigma-Aldrich, P4957) and 1 μg/mL fibronectin in proliferation medium. After 24 h, culture media was discarded, and fresh differentiation medium was added. After 48 h, cells were aggregated according to Neurix technology to generate air–liquid interface organoids called CellBrain. CellBrain organoids were kept for 3 days in differentiation media prior to testing, and thereafter the culture media were changed three times a week.

### 2.7. Biointegration as a Function of the Pressure

After autoclaving and 24 h before the experiment, the glymphatic brain-on-a-chip device was preconditioned. To do so, the reservoir and the collection tube were filled with LUHMES differentiation medium. The device was then transferred to the incubator set to 37 °C, 5% CO_2_, 95% humidity, and the pump was turned on for 24 h. For each condition, 2 mg of biomaterial was dried in a cell strainer (Greiner, 40 µm, 542040) placed onto a filtration unit (TPP, 191060) and then placed into a 6-well cell culture insert (Millicell^®^, pore size 0.44 um, Millipore, Burlington, MA, USA, PICM0350). On the top of each sample of dried biomaterial, a brain tissue surrogate was placed using a tweezer to carefully pinch only the PET membrane and not the tissue itself. The glymphatic brain-on-a-chip device was then placed in the incubator 37 °C, 5% CO_2_, 95% humidity and the pumps were turned on. The experiment was conducted for 7 days.

Biointegration scores were calculated as follows. Firstly, the height of the CellBrain was manually estimated using the FIJI software “Freehand lines” tool, from the bottom to the top based on the BIII-tubulin staining (averaging 5 to 10 measurements per cut). Using the same technique, the maximum distance between the top of the CellBrain and the biomaterial (biomaterial penetration), was assessed using the fluorescence of the biomaterial (thanks to 6-aminofluorescein). Finally, the ratio of averaged biomaterial penetration over averaged CellBrain height was then calculated and multiplied by 100 to give the biointegration score in percent.

### 2.8. Histology

After 7 days, brain tissue surrogates and the biomaterial were fixed by placing the 6-well culture insert in a 6-well plate with 1 mL of PFA 4%. They were then embedded into 2% agarose (*m*/*v*) before processing for dehydration (successive ethanol bathes, from 70% to 100%, *v*/*v*) then transferred to 50% ethanol and 50% xylol and then 100% xylol for 1 h. The samples were then embedded a mix of 2/3 paraffin and 1/3 ethylvinylacetate (EVA) as previously described [59]. After microtome cutting, cuts were dried overnight at 37 °C and then immerged sequentially in 2 hot baths of xylol (50 °C) for 15 min and processed for rehydration as usual paraffin cuts. The cuts were immersed in a blocking buffer (5% goat serum in 0.1% Tween). A primary antibody solution (BIII 1:1000, MAB1637, Millipore, Switzerland, in PBS) was added for 1 h at room temperature. Cleaved caspase-3 antibody (Cas-3 1:1000, 9661S, Cell Signaling, Danvers, MA, USA) was used for the viability assay. After 3 washings with PBS, the solution with the secondary antibodies and DAPI was added (1:1000, anti-mouse CF-680, Sigma-Aldrich, SAB4600361 and 1:1000, anti-rabbit CF-568, Sigma-Aldrich, SAB4600084 in PBS) for 1 h at room temperature. Finally, the cuts were washed 3 times with PBS and mounted for imaging on a LSM800 confocal microscope driven by Zen 3.1.00005 (blue edition, Zeiss) and using a 10× PlanApochromat lens with NA = 0.45.

For whole-mount analysis, the samples were also fixed in PFA 4% for 1 h. Then the samples were washed with PBS and incubated in blocking buffer (5% goat serum in 0.1% Tween) for 1 h. Samples were incubated overnight with BIII antibody (same condition/reference than mentioned previously) at 4 °C. After abundant washing with PBS, secondary antibody and DAPI were added for 2 h at room temperature. After washing with PBS, samples were loaded into an observation chamber (Marienfeld, Lauda-Königshofen, German, 1320102) and imaged using a LSM800 driven by Zen 3.1.00005 (blue edition, Zeiss) and using a 5× or 10× PlanApochromat lens with NA = 0.45.

### 2.9. hiPSCs 2D and 3D Culture

Maintenance and differentiation of neurons and glial cells from hiPSC neuro-progenitor cells was conducted based on a previously published protocol [60], with adaption for scaffold seeding. Briefly, HIP (A3890101, Thermo Fisher Scientific, Waltham, MA, USA) human neuro-progenitors derived from induced pluripotent stem cells were maintained in proliferation on 1:200 GelTrex LDEV-free hESC quality (A1413302, Thermo Fisher Scientific) coated flasks in expansion medium (Stempro™ NSC SFM kit, Thermo Fisher Scientific, A1050901) supplemented with 2 mM GlutaMAX ™ (Thermo Fisher Scientific, 35050038). For differentiation on scaffolds, cells were harvested using 2 mL of StemPro^TM^ Accutase^TM^ (Thermo Fisher Scientific, A1110501) and resuspended in a small volume of expansion medium in order to obtain a very dense cell suspension (ca. 200,000 cells per 10 µL). Meanwhile, sterile Matrigel-coated biomaterial, containing an equivalent of ca. 1 mg dry mass, was placed in cell strainer (40 µm mesh). Using a vacuum-driven filtration unit (TPP, 191060)), the material was washed several times using PBS in repeated dry-wash cycles. The use of a cell strainer is highly advantageous for small amounts of biomaterial in this step to minimize material loss. Then, the dried biomaterials were transferred onto a sterile hydrophilic PTFE membrane of tissue culture (6 mm diameter, laser cut, PTFE-001, HEPIA Biosciences) initially placed in a 6-well plate cell culture insert (Merck Millipore, Burlington, MA, USA, 044003). In order to ease the adhesion of the PTFE membrane onto the cell culture insert, the bottom of the well was filled with 1 mL of expansion medium. The cell culture insert membrane has to be in contact with the medium and completely wet prior to the transfer. The biomaterials were then seeded with 50,000 cells (i.e., 2.5 µL of the high density cell suspension). After 24 h in the expansion medium, cells were differentiated in a first differentiation medium (“DIFF1” [60]) composed of StemPro™ hESC SFM (Thermo Fisher Scientific, A1000701) completed with 20 ng/mL BDNF (Thermo Fisher Scientific, PHC7074), 20 ng/mL GDNF (Thermo Fisher Scientific, PHC7044), 500µM Dibutyryl cyclic AMP (Sigma, D0627) and 200 µM 2-phospho-Ascorbic Acid (Sigma, 49752). After one week, cells were differentiated in a second differentiation medium (“DIFF2” [60]) composed of DIFF1 medium and NDM medium (see below) at 1:1 ratio. After a further week of culture, cells were maintained with Neuron Differentiation and Maintenance Medium (“NDM” [60]) composed of B27™ Plus Neuronal Culture System (Thermo Fisher Scientific, Switzerland, A3653401) supplemented with 0.5 mM GlutaMAX ™ Supplement. Cultures were maintained at the air–liquid interface for 6 to 8 weeks in this medium. The medium was changed twice a week (1 mL) before transferring the PTFE membrane and the seeded biomaterial onto MEA Biochips for electrophysiology using sterile forceps, placing the biomaterial and its neural population in direct contact with the recording electrodes. Special care was taken to touch only the PTFE membrane in order to avoid direct contact of the forceps with the seeded biomaterial that could lead to undesired cell death.

### 2.10. Electrophysiology

Custom-made strip-micro-Electrode Arrays (MEA) were used to record the spontaneous activity of the neurons derived from human pluripotent stem cell (hiPSC). The manufactured strip-MEA devices are composed of an 8 µm thick porous polyimide membrane including four recording areas of eight recording sites each [61]. Electrodes have a diameter of 30 µm and are located on a 200 µm grid. Holes of diameter of 7.5 µm on a 20 µm grid etched through the membrane generate a 10% area membrane porosity at its workspace. The obtained polyimide membrane is connected to a printed circuit board allowing connection to signal amplification and acquisition electronics. The MEA device also includes fluidic channels for the culture medium that allowed us to apply similar culture condition as for the glymphatic brain-on-a-chip device (negative pressure and air–liquid interface) with a porous PET membrane acting as the permeable flow barrier.

To record the electrophysiological signals, we used signal amplifier and acquisition electronics commercially available by Multichannel Systems (MCS, Germany). Thirty-two electrodes at 20-kHz sampling rate record the electrical activity while the tissues on the strip-MEA are placed in an incubator under negative pressure. Data from the 32 electrodes were processed, transmitted by Wi-Fi and stored on a computer for further analysis. A/D conversion was done over 16 bits on a range of 25 mV. The commercial software from MCS was used to control the experiment as well as for basic processing/analysis of the raw data and spikes. Spike and timestamps detection were done using the thresholding method.

Once acquired, the raw data and waveforms were converted into hdf5 format and saved on the computer. Evaluation of the mean firing frequency of spikes and spike sorting were done using a custom-made graphical user interface (GUI). To objectively characterize the individual neurons, we used an algorithm implemented in the GUI, which is based on the signal-to-noise ratio (SNR) of the waveforms. This algorithm was used to classify the clusters as noise, multiple or single units. Noise units were ignored, whereas timestamps corresponding to multiple and single units were retained.

### 2.11. Mechanical Characterizations

Mechanical characterization was conducted on a TextureAnalyzer XTPlus machine from Stable Microsystems (United-Kingdom, temporarily replaced by a stepper-motor driven force sensor during COVID-19 access restrictions). A 4 mm cylindrical chuck was used for the compression testing of CellBrain samples, while in the case of cryogel biomaterial, bulk samples of 5 mm diameter, molded in 1 mL syringes, were used along with a larger chuck. For evaluation of the Young modulus, samples were compressed at a rate of 0.01 mm/s, and the modulus evaluated from the initial slope of the stress (force per area) vs. strain (deformation relative to original sample height). For stress relaxation experiments, the samples were compressed by about 50% of their original height (as determined from the onset of detectable elastic recoil force), and the deformation maintained for 1 h. The evolution of the Young modulus over time during stress relaxation experiments was estimated by relating effective stress (after subtraction of baseline stress due to buoyancy and capillarity effects) to the imposed strain.

## 3. Results

### 3.1. Results

#### 3.1.1. Glymphatic Brain-on-a-Chip Principle and Characterization

Figure 1 outlines the engineering strategy in the design of a glymphatic-brain-on-a chip culture device. The aim is to mimic essential physico–chemical characteristics of the physiologic glymphatic system. First, in the native glymphatic system (Figure 1a), relatively rapid convective flow occurs in the perivascular space, driven by the arteriolar pulsatility and respiration [45], whereas the astroglial blood–brain barrier constitutes a very high fluidic resistance, permitting only minimal convective flow in the interstitial space [62]. Second, despite high fluidic resistance, the glymphatic system is capable of high metabolite and blood gas exchange rate, due to short diffusion distances [45]. Third, the physiological filtration equilibrium imposes a negative IFP in the range of −0.4 to −1.5 kPa (average 0.9 ± 0.4 kPa) [44]. Thus, the glymphatic brain-on-a-chip should recapitulate the important flow rate differential between the perivascular space and the neural interstitium, while permitting rapid metabolite exchange. Crucially, it should allow the IFP to set in a controlled manner.

In the glymphatic brain-on-a-chip system, a capillary bridge allows for rapid conduction of fluids (Figure 1b) mimicking perivascular space of the brain (Figure 1a). In our glymphatic model system, we used cell culture insert membranes (Figure 1b, pore diameter: 0.44 µm) to shield the CellBrain environment from the capillary flow, while permitting diffusion of nutrients and waste products. The cell culture insert membrane therefore mimics the fluidic role of the astrocyte barrier. For gas exchange, we mimicked the presence of blood-carrying capillaries by proximity to a free air–liquid interface on top of the neural culture area.

Figure 2 shows the physical implementation of the glymphatic brain-on-a-chip culture system. An external pump ensures persistence of a hydrostatic pressure difference between a reservoir tube and a collector tube (Figure 2a). This emulates the fluid driving force that arterial pulsation is known to exert on the paravascular glymphatic conduits [63]. The externally imposed pressure difference indeed leads to constant capillary flow in the capillary bridge of the glymphatic brain-on-chip device (Figure 2a,b) similarly to the flow in the paravascular conduits (Figure 1a). In vivo, the negative IFP is thought to arise by a difference in oncotic pressure between IF and blood plasma as well as lymphatic fluid uptake [35,64].

The cell culture insert membranes lack molecular selectivity and therefore cannot produce an oncotic pressure difference. We therefore impose the negative IFP directly, by capillary contiguity (Figure 2a). The device achieves this based on the principle of communicating vessels, as well as asymmetric design of the capillary bridge (Figure 2a, yellow arrow). This capillary bridge is indeed fabricated with a wide end connected to the reservoir tube and a narrow end connected to the collector tube. The wide end allows free transmission of the negative pressure arising from the height difference between reservoir and glymphatic brain-on-a-chip area, while the narrow end acts as a slow drain to the collector tube and therefore as a flow regulator. In fine, the pressure applied to the sample will be determined almost exclusively by the difference in height between the sample and the reservoir, while the flow rate can be controlled by the pressure difference between reservoir and collector tube.

To extend the glymphatic brain-on-a-chip principle to microelectrode-based neural recording, we adapted flexible polyimide-based microelectrode array technology (Figure 2c,d). Specifically, we etched vias of 7.5 μm diameter through flexible microelectrode arrays, ensuring capillary contiguity, and combined the recording setup with a capillary fluidic bridge able to expose the CellBrain to a known and defined negative pressure during culture and recording episodes.

In order to validate the pressure applied to the samples, we measured the pressure in the capillary bridge using a custom pressure measurement device. As expected, the values measured were similar to the ones we theoretically defined (slope = 0.999 ± 0.015, intercept = −4.25 ± 7.14 Pa, *p* = 0.929, for comparison against a theoretical slope of 1 in linear regression). Moreover, we also measured the typical range of negative pressures in an air–liquid interface culture using a cell culture insert. We found that the average pressure applied for cell culture insert was around −0.075 kPa. These results demonstrate that the glymphatic brain-on-a-chip can accurately apply a wide range of well-controlled negative pressure. Most importantly, and contrary to established cell culture insert technology, the glymphatic brain-on-a-chip is able to cover the physiological IFP range in the brain.

#### 3.1.2. Biomaterial Fabrication and Covalent Attachment of Cell-Adhesive Proteins

With device fabrication and characterization complete, we next proceeded to evaluation of neural biomaterial integration as a function of IFP. We are particularly interested in biointegration of porous biomaterials. These materials have indeed an outstanding capacity to support colonization of the pore space in vivo, particularly if the pores are sufficiently large to support vascularization [24,65]. To assess the effect of IFP on biointegration, we used a porous sponge-like biomaterial based on carboxymethylcellulose, synthesized by cryogelation and fragmentation [25,55].

To quantify the impact of adhesion motives along with the one of IFP, we covalently modified the biomaterial with ECM proteins using carbodiimide chemistry [56], as outlined in Figure 3a. We found the coating to be relatively homogeneous, even if there are occasional protein clusters (Figure 3b vs. Figure 3c, Matrigel). In order to assess the efficiency and stability of the covalent protein attachment, we quantified the amount of proteins on the biomaterial remaining after 24 h at 37 °C. We did so for the nominal modification procedure for Matrigel, but also for negative controls without EDC and also without ECM protein (Figure 3d). By using the Bradford protein quantification assay, we detected around 89.55 ± 36.23 µg protein/mg CMC after EDC treatment, while only 11.88 ± 6.54 µg/mg CMC was detectable in the absence of EDC treatment (*p* = 0.003, *t*-test with Bonferroni correction). Virtually no protein was detected when no ECM protein was provided (1.95 ± 1.55 µg/mg CMC). This confirms the specificity of the assay and further indicates that covalent attachment of Matrigel protein components onto the surface of the biomaterial dramatically enhances ECM protein retention. As the initial adsorption step in the protocols with and without EDC are identical, we concluded that EDC-mediated amide bond formation is necessary to avoid subsequent protein desorption. Covalent attachment therefore is needed to ensure biomaterial stability during longer-term in vitro cell culture, but also simply biomaterial storage.

#### 3.1.3. Interstitial Fluid Pressure-Dependent Biointegration by a Porous Biomaterial Scaffold

We next sought to investigate the integration of the porous biomaterial with the brain tissue surrogate under negative pressure conditions. To this end, we cultured for 7 days the neural organoids sold under the commercial name of CellBrain, a tissue-like assembly of NSC-based neurospheres (LUHMES cells) in contact with the porous biomaterial. We used the IFP device to apply different negative pressure values (Figure 3a): low (−0.3 kPa), moderate (−0.6 kPa) and high (−0.9 kPa). In this selection, −0.3 kPa is expected to not reach the physiological range, while −0.6 and −0.9 kPa are similar to the physiological values observed by Brodersen et al. [44]. For each condition, we used the porous biomaterial unmodified and modified either with collagen type I, Matrigel or laminin. We then quantified biointegration as the ratio between biomaterial penetration into the CellBrain and the CellBrain average thickness.

At low negative pressure, namely −0.3 kPa, poor biomaterial integration was observed regardless of the protein modifications (Figure 4b, all *p* > 0.2, one-way ANOVA, as compared to no coating/collagen). For both moderately (Figure 4c) and highly negative pressure (Figure 4d, −0.6 kPa and −0.9 kPa), biomaterial integration was greatly improved for all coating conditions (*p* < 0.002 within each coating condition, linear regression on IFP). In addition, we observed significantly enhanced biointegration for laminin and Matrigel as compared to collagen in the moderate and high negative IFP conditions (*p* ≤ 0.03 for both −0.6 and −0.9 kPa, one-way ANOVA, as compared to no coating).

To better understand the respective contributions of IFP and coating to biointegration, we conducted a two-way ANOVA with both the magnitude of the IFP and the type of coating as explanatory variables for the biointegration score. This analysis reveals that while both IFP and coating contribute highly significantly (P_pressure_ = 3.7 × 10^−18^ vs. P_coating_ = 4.3 × 10^−7^ following overall significant ANOVA, P_ANOVA_ = 2.9 × 10^−16^), the IFP explains a substantially greater fraction of the variance (59.4% for the IFP vs. 16.6% for the coating). Thus, negative IFP has a central role in the in vitro integration of the biomaterial into an established brain tissue surrogate. As evident from Figure 4, there is a threshold effect, biointegration being poor at −0.3 kPa regardless of the coating, with a step improvement to −0.6 kPa, but no further significant difference to −0.9 kPa. As anticipated [57], laminin and Matrigel coatings do have a favorable effect on biointegration, albeit for the bulk tissue invasion investigated here, this seems to depend on reaching the necessary IFP threshold.

Finally, it is noteworthy that the higher negative IFP values (−0.6 kPa and −0.9 kPa) not only increase the colonized area, but also the cell density in the colonized area as compared to −0.3 kPa (Appendix A). Indeed, averaged across the different coating conditions, at −0.3 kPa, a volumetric cell density of 150 ± 70 million cells is reached in the colonized area. This number rises to 420 ± 90 million cells/mL at −0.6 kPa (*p* = 0.045 vs. −0.3 kPa) with no further significant change to −0.9 kPa (*p* = 0.61 vs. −0.6 kPa) while the significant difference vs. −0.3 kPa is maintained (*p* = 0.0078 vs. −0.9 kPa vs. −0.3 kPa). No significant influence of coating on cell density could be found (Appendix A). In absolute terms, these cell densities achieved are very high: they are reminiscent of solid tissue [66] and thus suggest volumetric organoid bulk colonization, particularly in the case of the higher IFP values.

#### 3.1.4. Spontaneous Electrical Activity of hiPSCs 3D Culture under Negative Pressure Conditions

We then investigated the impact of the glymphatic brain-on-a-chip culture condition (air–liquid interface and negative pressure) on the functionality of 3D culture hiPSCs-derived neurons. First, we investigated cell viability via cleaved caspase-3 staining and found no ill effects (Appendix A). Next, we investigated electrical activity under negative pressure conditions, by the custom Biochip presented in Figure 2c,d. This system allowed for electrical activity measurement of the hiPSCs-derived neurons previously differentiated for 8 weeks in Matrigel-modified biomaterials (Figure 5a), while maintaining the proximity of the air–liquid interface and application of known IFP.

Figure 5b shows a representative spontaneous activity recorded from 2 electrodes of 1 strip-MEA. Typically, the intensity and the frequency of the signal were not homogenously distributed among all the 8 MEA electrodes (also shown in Figure 4c, after thresholding) due to random placement of the biomaterial-neuro-organoid hybrids. As shown in Figure 5d, it is nevertheless possible to detect and follow action potentials of a single neuron.

Altogether, these results show that neurons derived from hiPSCs are capable of spontaneous electrical activity in similar culture conditions provided by the brain-on-a-chip device and that detection is possible in this ultimately 3D geometry.

#### 3.1.5. Mechanical Characterization of the CellBrains

Our results suggest that negative fluid pressure acts as a major factor favoring biointegration. To better understand the mechanism behind this effect, we undertook mechanical characterization of both the CellBrain and the biomaterial.

Our first step was to check the validity of the CellBrain model to reproduce the mechanical brain tissue behavior. For this, we measured the stiffness of the brain organoid by uniaxial compression and we found an average Young’s modulus of 3.5 ± 1.9 kPa. This is similar to literature values (between 1.5 and 2 kPa) of the brain established with similar measurement techniques [67]. Thus, these results strongly suggest that the CellBrain tissue surrogate is a suitable in vitro model to study the mechanical interaction between biomaterials and the brain tissue.

Next, we characterized bulk samples of the biomaterial under similar conditions. Here, we found a Young’s modulus of 1.3 ± 0.6 kPa, in agreement with previous measurements [25]. This is comparable with native brain tissue moduli [67] and with the CellBrain model. This result indicates that our biomaterial is suitable to avoid mechanical mismatch with the brain. It would therefore constitute a mechanically suitable vehicle for either cell transplantation or bulking [55,57]. But this finding does not provide an explanation or mechanism for why biointegration would occur with the CellBrain model under negative IFP. In fact, it is important to note that in vivo biointegration is a slow, progressive phenomenon [25]. Furthermore, native tissues have been reported to have very strongly time-dependent mechanical characteristics [68,69].

We therefore specifically compared the time-dependent mechanical response of the CellBrain on the one hand, and the porous biomaterial on the other. We examined stress relaxation for this purpose. In stress relaxation, a given deformation is imposed on the material, and the ensuing stress is followed over time. While the Young moduli alone would suggest very similar mechanical behavior in the CellBrain and the biomaterial samples, stress relaxation indicates a striking difference: Very little stress relaxation is seen for the biomaterial, while the CellBrain tissue surrogate lost over 90% of its initial Young’s modulus in 10 min (Figure 6b,c, CellBrain vs. biomaterial: *p* = 0.03, *t*-test of relative loss of stress). This means that at even relatively short time points, the CellBrain tissue surrogate is unable to sustain forces that can be handled easily by the biomaterial and will ultimately be subjected to large-scale deformation despite an apparently higher initial Young’s modulus. Taken together, these results mean that the biomaterial behaves as a stable, elastic material, while the CellBrain tissue has elastic properties mostly in the short term, while exhibiting the potential for plastic (viscous) deformation in the longer term. Together with the culture experiments on the glymphatic brain-on-chip platform, it appears that at least the CellBrain tissue surrogate survives such slow plastic deformation without further harm, adapting progressively to its novel geometry.

## 4. Discussion

In this study, we present an in vitro approach able to mimic physiological (g)lymphatic pressure of tissues in order to recapitulate the physiological forces applied on a porous biomaterial upon transplantation. This was achieved by the unique combination of our novel hydrostatic glymphatic brain-on-a-chip pressure and culture device and an already established and commercially available model of brain tissue surrogate, the CellBrain.

According to characterization by uniaxial compression, the CellBrain tissue surrogate has shown a similar range of stiffness than previously described for brain tissues with similar techniques [67]. This result supports the use of the CellBrain as a suitable in vitro model to investigate the mechanical interactions between the biomaterial and the brain tissue. Thereafter, we used the system to study the role of IFP in comparison to established ECM modification to foster integration of a porous CMC-based biomaterial into the surrogate brain tissue in vitro. First, as one may anticipate [57], we found that cell affinity to the biomaterial favors biomaterial integration, although at low negative IFP conditions, integration remains inefficient and the effect of ECM proteins minor. This probably reflects the known fact that neurons at an advanced stage of maturation, such as present in the CellBrain [58], have limited migration capacities [70]. The observed substrate effect might therefore be resulting from progenitor migration or neurite sprouting, particularly on substrates such as laminin or Matrigel [70,71,72,73].

Instead, we found that the in vitro tissue integration was mainly pressure-dependent and maximal at the physiological values (−0.9 kPa for the brain [44]). The negative pressure apparently exerts sufficient forces on the CellBrain tissue to be slowly drawn into the biomaterial, including in the total absence of cell-adhesive coatings. Ultimately, the biomaterial becomes intermingled with the tissue surrogate, providing colonization with cell densities in the hundreds of millions per mL range akin to solid tissue [66].

At first glance, it may seem difficult to reconcile the very efficient 3D migration bulk migration observed under the persistent influence of negative IFP with the known incompatibility of mature neurons with migration [70]. To better understand the process, we conducted a detailed time-dependent characterization of the mechanical response of both the biomaterial and the CellBrain nervous tissue. In the short-term, according to relatively rapid uniaxial compression measurements, it would appear that the CellBrain tissue is comparable, or even slightly stiffer, than the biomaterial. However, analysis of stress relaxation reveals a dramatic difference: while hardly any stress relaxation is observed for the biomaterial, the CellBrain tissue loses over 90% of its stiffness in just a few minutes. This means that under low, but constant forces such as the ones produced by persistent negative pressure, the CellBrain tissue will resist elastically only for a short time, before being subject to progressive fluid-like mechanical deformation. Thanks to this plastic deformation, it is slowly squeezed into the pore space, although the process happens at a pace slow enough for the cells to progressively adapt. We observed thus not so much cell migration *per se*, but a form of bulk tissue migration. Our data further suggest that this bulk deformation and migration is not accompanied by a loss of cell viability and therefore indicates a novel pathway to producing completely integrated 3D biomaterial-tissue hybrids. If prestructured 3D tissues are grown, it would be possible to produce biomaterial reinforced, surgically handleable minitissues.

Even though our results are at present purely based on an in vitro model, they suggest that in situ tissue integration might be strongly driven by biomaterial geometry and physical properties, surface modification playing possibly only a minor role in this regard, in particular for the brain tissue. This somewhat challenges the role of the use of cell-adhesion motives [27,28,29,30] as main constituents or to decorate biomaterial surfaces, at least in systems with relatively poor cell migration ability. In such cases, the specific biomaterial or its interaction with cells may not be all-important for biointegration. Chemical modification of biomaterials serves various purposes such as delivery of adherent cells, immunomodulation, chemiotaxis or the delivery of specific biological signals [57,74,75], but the slow process of tissue deformation by physical forces seems to be able to provide colonization efficiencies that the migratory properties of the individual cells would never have suggested. If indeed exploitable in vivo, slow integration of endogenous brain tissues could offer fundamentally new therapeutic concepts: in stroke, one could for instance envision scaffold integration beyond the lesion core to provide a guided path connecting healthy tissue points; in cell transplantation, one could willfully target intermingling of engineered implants with existing structures, both for targeted reconstruction of neural pathways and efficient metabolic supply. The source of negative pressure could be physiological processes [44], with possible enhancement through drainage [76] or through biological lymphatic engineering [77,78]. Whatever the modalities to come, this report lays a rational basis for corresponding biomaterial design and parameter testing.

Beyond a clear application for the brain, our hydrostatic device is able to reproduce a wide range of IFP, matching other tissues (lungs: −0.7 kPa, muscles: −0.3 kPa, subcutaneous tissues: −0.8 kPa [79]). Hence, other tissue surrogates might be used (for instance tissue-engineered human skin equivalents [80]) to predict and optimize tissue pressure-depend response of the biomaterial before their transplantation in vivo. This approach is rather inexpensive and could provide a reliable screening platform to test and optimize biomaterial stiffness and mechanical response in physiological conditions while reducing the need for animal research.

The scope of the biophysical role of the IFP in biointegration in vivo remains to be determined. However, if interstitial fluid pressure plays an important role in biointegration also in vivo, it could offer a novel perspective on a series of phenomena such as the known negative impact of inflammation on biomaterial colonization despite a plethora of chemoattractant molecules secreted by immune cells, and more generally, give an unanticipated dimension to the (g)lymphatic colonization of implants beyond the well-known impact of proper vascularization.

## 5. Conclusions

Our data are in line with the idea that implant chemistry is only part of the key to success in biomaterial integration—another important part is linked to mechanical properties and geometric structure. As such, the porous biomaterial described in this study seems promising for brain applications, as with the known strongly negative IFP in the brain one may anticipate a strongly favorable effect on potential biointegration, possibly including neural elements. On the fundamental level, our results propose a novel, highly effective tissue integration mechanism based on long-term viscoelastic bulk tissue deformation. At the heart of the type of biointegration is the interplay between interstitial fluid pressure and stress relaxation—both are aspects that in light of the results presented here have received too little attention so far.

## Figures and Tables

**Figure 1 brainsci-12-00417-f001:**
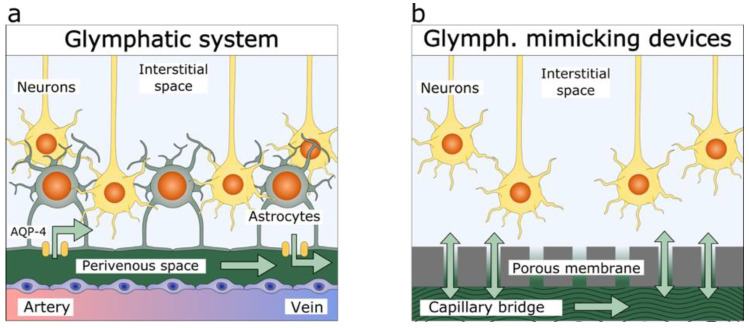
Comparison of the (**a**) glymphatic system and (**b**) glymphatic brain-on-a-chip. Green arrows show the fluid flow in both systems. (**a**) Interstitial fluid exchange is possible through the blood–brain barrier. The glymphatic system has a given selectivity thanks to the presence of aquaporine-4, (AQP-4). (**b**) Unlike the glymphatic system, the membrane of the glymphatic brain-on-a-chip is permeable to all solutes. The system allows the free exchange of fluids and the culture of brain organoids at the air–liquid interface.

**Figure 2 brainsci-12-00417-f002:**
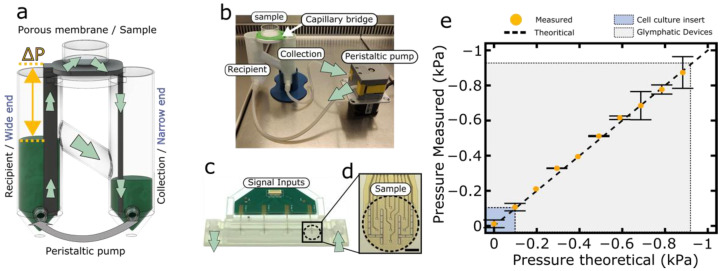
Glymphatic brain-on-a-chip devices and characterization. (**a**) Schematic representation of the glymphatic brain-on-a-chip. It is composed of a recipient and a collection tube, a capillary bridge and a peristaltic pump. The green arrows represent the fluid flow. A hole in the recipient tube at a defined height (3, 6 or 9 cm) allows excess fluid to pass to the collection tube and to apply a controlled negative pressure on the sample (yellow arrows, respectively −0.3, −0.6 or −0.9 kPa). (**b**) Picture of the experimental set-up with 3D-printed Recipient/collection tube. (**c**) Picture of a Biochip showing 4 strip-MicroElectrode Arrays (MEA) on top of a Polyethylene terephthalate (PET) porous membrane. An intermediate printed circuit board (PCB) allows the connection of the 4 strip-MEAs to external signal amplification and recording electronics. (**d**) Zoom around the tip of a microfabricated strip-MEA showing 8 platinum recording electrodes as well as two ground electrodes. The PET porous membrane allows for the application of a known negative pressure and air–liquid interface culture. This measurement set-up may be added on the glymphatic brain-on-a-chip to perform in-real time spontaneous neuronal activity. Scale bar = 200 µm. (**e**) Theoretical/measured pressured curve showing the range of pressure achieved by the Glymphatic brain-on-a-chip (grey box) as compared to the air–liquid culture performed in a regular cell culture insert (blue box).

**Figure 3 brainsci-12-00417-f003:**
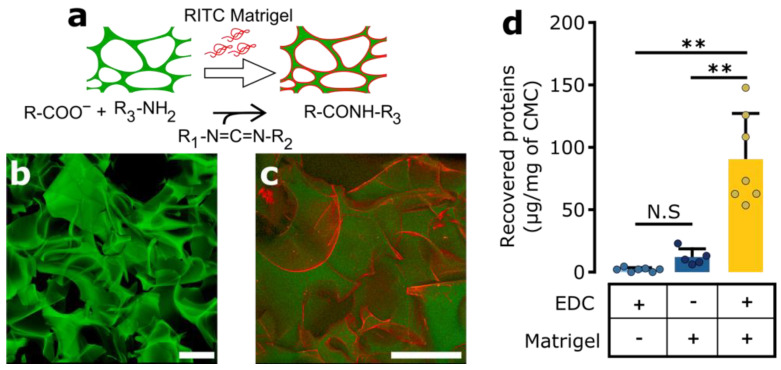
Porous biomaterial scaffold structural view and protein covalent post-modification. (**a**) Principle of the EDC (1-ethyl-3-(-(3-dimethylaminopropyl) carbodiimide) protein covalent modification. Covalent protein attachment on the carboxylmethylcellulose (CMC) using amide bond formation is mediated by application of a water-soluble carbodiimide, EDC. (**b**) Microscopic view of the porous biomaterial (naive scaffold, synthesized with 6-aminofluorescein). (**c**) Visualization of protein attachment directly on the biomaterial by RITC staining. After covalent protein modification, the Matrigel is attached to the surface forming a relatively uniform layer. (**d**) Quantification of the protein immobilization. Matrigel (initially 1 mg Matrigel/mg CMC) was immobilized (various conditions, with or without protein covalent immobilization with EDC) followed by 24 h of incubation in DMEM, at 37 °C. The remaining protein content was recovered from the surface by enzymatic treatment and assessed using Bradford technique (expressed in µg/mg of CMC). *n* = 4–5. Statistical analysis performed using t-test and Bonferroni multiple testing correction. Scale bars = 200 µm. (**: *p* < 0.01).

**Figure 4 brainsci-12-00417-f004:**
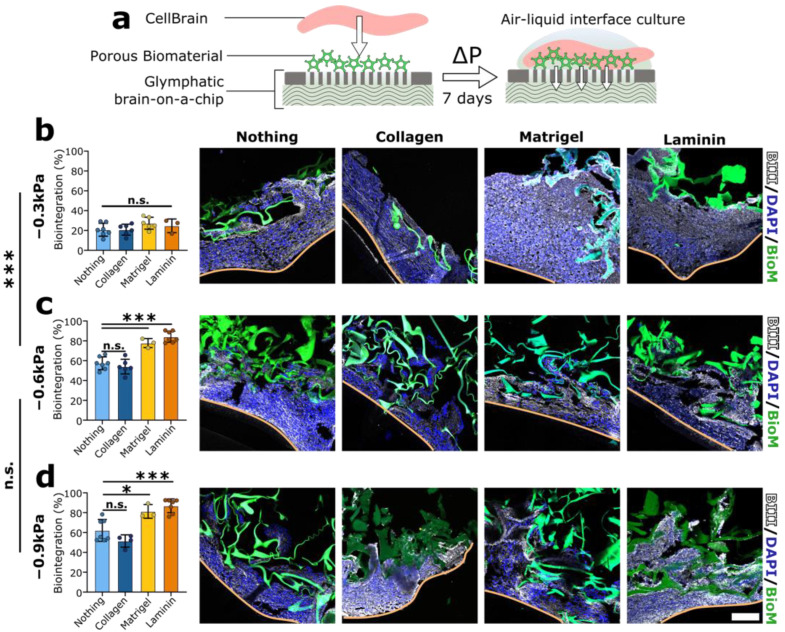
In vitro pressure dependent biointegration using the glymphatic brain-on-a-chip device. (**a**) Workflow. The CellBrain and the porous biomaterials post-modified with EDC only (“Nothing”, no ECM) or with different ECM proteins (namely collagen I, Matrigel and laminin) were directly in contact with the membrane (0.44 µm) and the capillary bridge of the glymphatic brain-on-a-chip device. A range of negative pressures (from −0.3 to −0.9 kPa) was applied for 7 days. (**b**–**d**) Quantification and representative pictures of the in vitro biointegration at −0.3 (not physiological IFP) −0.6 and −0.9 kPa (falling withing the physiological range of IFP). (**b**) The integration of the biomaterial was minor with no significant differences between the ECM proteins. (**c**,**d**) The biointegration was greatly improved for all the ECM protein conditions at more negative IFP as compared to −0.3 kPa. Enhanced biointegration specifically attributed to Matrigel and laminin was observed but remains minor compared to the effect of the pressure. BIII-tubulin (BIII) immunofluorescence, in white, was performed to visualize the CellBrain. Biomaterial (BioM), in green, was synthesized with 6-aminofluorescein, while nuclei were visualized in blue using DAPI. *n* = 3–7. Three distinct experimental replicates. One-way ANOVA statistical analysis. The orange lines the side of the CellBrain at the air–liquid interface. Scale bar = 100 µm. ( n.s.: *p* > 0.05; *: *p* < 0.05; ***: *p* < 0.001).

**Figure 5 brainsci-12-00417-f005:**
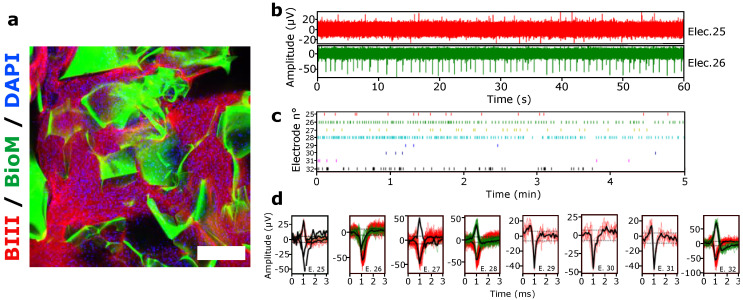
3D Electrophysiological measurements of hiPSCs-derived neurons under negative pressure conditions. (**a**) hiPSCs-derived neurons cultured and differentiated in 3D in the porous biomaterial scaffold for 8 weeks. These 3D cultures were then transferred into the strip-Biochip for electrophysiological measurement. (**b**) Typical time series of spontaneous activity recorded from 2 electrodes (#25 in red and #26 in green). (**c**) Timestamps indicating biological events detected in real time by thresholding (±6 standard deviation of the noise). (**d**) Detected action potentials of single neurons. Neurons with firing frequency ranging from 0.47 Hz to 2.22 Hz were typically detected. Scale bars = 200 µm.

**Figure 6 brainsci-12-00417-f006:**
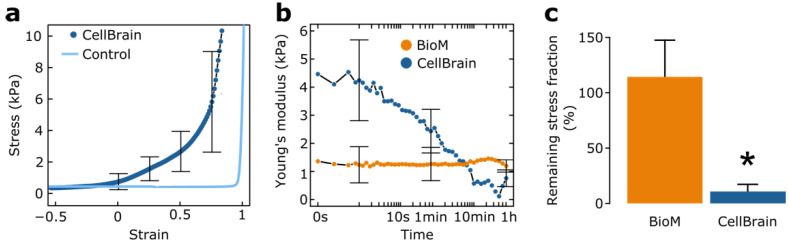
Mechanical characterization of the brain tissue surrogate CellBrain. (**a**) Uniaxial compression of CellBrain (blue line) and control (PET membrane used as a support for the CellBrain culture, orange line). The Young’s modulus of the CellBrain is 3.5+/−1.9 kPa. *n* = 5. (**b**) Stress relaxation of CellBrain (dark blue line) and the porous biomaterial (BioM, orange line) as a bulk (before fragmentation). While the Young’s modulus of the CellBrain rapidly decreases over time (around 90% within few minutes), the Young’s modulus of the porous biomaterial remains constant. *n* = 4 for the CellBrain and *n* = 3 for the biomaterial. (**c**) Remaining stress fraction in % (ratio of the stress relaxation at *t* = 0 s over *t* = 1 h) for the biomaterial (BioM, orange) and CellBrain (dark blue). Statistical analysis: *t*-test on the ratio between early (<5 s) and late (>1800 s) stress, CellBrain vs. biomaterial. (*: *p* < 0.05).

## Data Availability

Raw data are available online at http://doi.org/10.5281/zenodo.5115814. A Codeocean capsule (Capsule 7598976, https://doi.org/10.24433/CO.7598976.v1) is also available to allow automatic plotting of the manuscript figures. Custom R libraries used for data evaluation and plotting are available at the Zenodo repository as well as live on Github (textureAnalyzerGels: https://doi.org/10.5281/zenodo.4589276 respectively https://github.com/tbgitoo/textureAnalyzerGels, plot.counts: https://doi.org/10.5281/zenodo.4589498 respectively https://github.com/tbgitoo/plot.counts, reproducibleCalculationTools: https://doi.org/10.5281/zenodo.4594515 respectively https://github.com/tbgitoo/reproducibleCalculationTools) (accessed on 2 February 2022).

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
