# Peer review of "The Role of Interstitial Fluid Pressure in Cerebral Porous Biomaterial Integration"

_brainsci, 2022, doi:10.3390/brainsci12040417_

Round 1
Reviewer 1 Report
The manuscript represents a well design study investigating the role of interstitial fluid pressure for the integration of biomaterial. It highlights the importance of interstitial fluid pressure that which has not been addressed thoroughly previously.
It is written in a concise language which properly describes the material. The study is well conducted and presents the original work of authors where they prototyped a brain-on-a chip device that fulfilled the purpose of the study. The results of the study are novel, interesting and important for further clinical implications.
The methods described represent an extensive original contribution of authors and are the major strength of the study. In the methods and results section, a slight confusion has been crated, since both sections describe intermingled information of both methods and results. It would propose that more clarified distinction is made between both sections.
The discussion summarizes the results properly. In the discussion section a short description of the possible clinical implications would be welcome, since it would expand the potential of the study.
All together, the study presents a valuable novel insight and I would recommend it to be published.
Author Response
We wish to thank here the reviewer for the time and care taken to assess this manuscript and the constructive comments.
Reviewer#1 comment 1: The methods described represent an extensive original contribution of authors and are the major strength of the study. In the methods and results section, a slight confusion has been crated, since both sections describe intermingled information of both methods and results. It would propose that more clarified distinction is made between both sections.
Our reply: There is always a certain compromise between referring in a detailed manner to the methods used and avoiding redundancy between the Results and the Methods section. As the reviewer states, one of the major strengths of the paper is its methods, and we felt that it was important to let the readers be able to follow what methods were used with precision in the results section. We have nevertheless looked carefully through the manuscript and identified a few elements that could be stated more precisely to avoid redundancy and also relocated minor methodological information given in the Results section to the Methods section. In detail:
In section 3.1.2, lines 416-422 in the original version, minor methodological details about biomaterial synthesis and covalent modification were given. We moved these details to the Methods section and now simply state in the Results section that the biomaterial is porous and sponge-like and is obtained by cryogelation and fragmentation; likewise, the EDC chemistry is now mentioned only for identification than explained in the results section. Correspondingly, the details about amidation and synthesis of bulk scaffolds followed by fragmentation are now completely in the methods section. Likewise, we moved the detail about incubation in DMEM for evaluation of protein loss to the methods section (line 423).
In section 3.1.3, on line 448 in the original version, the staining details for quantification of the biointegration were given (BIII-tubulin and aminofluorescein); this information is indeed completely redundant with the Methods section and figure caption of figure 4 and so we removed it for conciseness from the Results section.
In section 3.1.4, the methodological detail of using a porous PET membrane as a flow barrier when performing electrophysiology was indicated only in the results section (line 490), we now moved this technical detail to the methods section.
Finally, the title of section 3.1.3 was placed a bit too far down in the manuscript, we moved it to its correct place.
We hope that these simplifications and clarifications help increase readability of the Results section, while assembling all pertinent methodological details in the Methods section.
Reviewer#1 comment 2: The discussion summarizes the results properly. In the discussion section a short description of the possible clinical implications would be welcome, since it would expand the potential of the study.
Our reply: This is a very constructive suggestion. There are indeed interesting clinical perspectives, possibly quite radically novel. One could for instance imagine to use negative pressure-driven biointegration in stroke to achieve biomaterial integration not only throughout the lesion, but out into healthy tissue. In this case, one could imagine providing a path for regeneration through the lesion, but from healthy tissue to healthy tissue, and could thus help to address the problem of lesional walling off. Another interesting application could be cell transplantation with subsequent mechanical integration of the transplant with the surrounding tissue, again providing a contact that more traditional methods do not deliver. We now mention these outlooks in the discussion.
Reviewer 2 Report
This is a very novel and thorough study. I have only a few suggestions.
The authors have left out important information pertaining to the culture conditions of iPSCs and associated neuronal differentiation. The authors only include the following statements:
"Maintenance and differentiation of the hiPSC has been describe previously[59]."
"This system allows for electrical activity measurement of the hiPSCs derived neurons previously differentiated for 8 weeks in Matrigel-modified biomaterials (Figure 4a)."
The differentiation of neurons from iPSCs is not standardized - certainly not using scaffold-based cultures. There are a variety of methods that can be used for these types of cultures and knowing the specifics of these methods is essential - especially considering that the resulting iPSC-derived neurons are not extensively characterized.
In Figure 5, the beta III tubulin expression and neuronal morphology is somewhat difficult to distinguish as the pseudocolor of the immunostaining is white. I would recommend changing this pseudocolor to something more commonly used (e.g. red) that better offsets the colors of the scaffold and DAPI.
Lastly, there are multiple spelling and grammar errors that should be corrected.
Author Response
Reviewer#2: This is a very novel and thorough study. I have only a few suggestions.
Our reply: We thank the reviewer for the effort made to review this manuscript and the valuable suggestions.
Reviewer#2 comment 1: The authors have left out important information pertaining to the culture conditions of iPSCs and associated neuronal differentiation. The authors only include the following statements:
"Maintenance and differentiation of the hiPSC has been describe previously[59]."
"This system allows for electrical activity measurement of the hiPSCs derived neurons previously differentiated for 8 weeks in Matrigel-modified biomaterials (Figure 4a)."
The differentiation of neurons from iPSCs is not standardized - certainly not using scaffold-based cultures. There are a variety of methods that can be used for these types of cultures and knowing the specifics of these methods is essential - especially considering that the resulting iPSC-derived neurons are not extensively characterized.
Our reply: We are grateful for this remark, it is true that even though the general outlines of the neural differentiation are known, it is important for the reader to be able to understand and eventually be able to replicate the detailed protocol. We therefore now provide a paragraph in the section Materials and Methods (current version paragraph 2.9 iPSCs 2D and 3D culture, line 291) that describes thoroughly the 2D expansion as well as the 3D culture conditions and transfer to the electrophysiology Biochip.
Reviewer#2 comment 2: In Figure 5, the beta III tubulin expression and neuronal morphology is somewhat difficult to distinguish as the pseudocolor of the immunostaining is white. I would recommend changing this pseudocolor to something more commonly used (e.g. red) that better offsets the colors of the scaffold and DAPI.
Our reply: We agree with reviewer#2 and have changed the pseudocolor from white to red for figure 5.
Reviewer#2 comment 3: Lastly, there are multiple spelling and grammar errors that should be corrected.
Our reply: Yes, we tried to review the manuscript thoroughly for spelling and grammar errors.